# Impact of Self-Control and Social Network of Friends on the Amount of Smoking among Out-of-School Youth

**DOI:** 10.3390/healthcare10112138

**Published:** 2022-10-27

**Authors:** MinHee Park, HyeYoung Song

**Affiliations:** 1Department of Nursing, Wonkwang University, 460 Iksandae-ro, Iksan-si 55338, Jeonbuk, Korea; 2College of Nursing, Woosuk University, 443 Samnye-ro, Samnye-eup, Wanju-gun 55338, Jeonbuk, Korea

**Keywords:** self-control, social network, friends, smoking

## Abstract

The aim of this study is to understand the effects of self-control and social networks of friends on the amount of smoking among out-of-school adolescents. The subjects of this study were 187 out-of-school adolescent smokers from the J-province Youth Counseling Welfare Center as of 2020. Data were collected through self-report questionnaires that contained questions about sociodemographic characteristics, self-control, social networks of friends, and amount of smoking. The dependent variable was smoking amount. Descriptive statistics, χ^2^ tests, correlation analyses, and regression analysis were performed. The predictors of smoking in OSY (out-of-school youth) were analyzed with respect to self-control and social networks of friends. The significant variables in model 3 were age, living with parents, and average allowance. The smoking rate of friends (β = 0.256) and the degree of penetration of friends smoking (β = 0.341) were significant variables. The higher the percentage of friends smoking and the higher the degree of penetration of smoking among the members of social networks, the higher the amount of smoking.

## 1. Introduction

Out-of-school youth are adolescents aged between 9 and 24 who have been absent from elementary or middle school for more than three months, have been on probation, dropped out of high school, or did not enter high school according to Article 2 of “Korea’s Act on the support for out-of-school youth”, meaning that they have given up studying prior to completion of their education. Tobacco is considered a harmful drug according to “Korea’s Adolescent Protection Act”; therefore, it should not be sold to youth. Voluntary efforts should be made to protect adolescents from tobacco, such as by preventing the distribution of cigarettes to youth.

According to a survey of out-of-school youth conducted by the Ministry of Gender Equality and Family and Korea Youth Policy Institute (2018) [1], the prevalence of smoking among out-of-school adolescents was 62.4% (70% of male out-of-school adolescents and 47.7% of female out-of-school adolescents were smokers). This result is significantly higher than the prevalence of smoking among adolescents attending school of 6.7% (9.4% for male students and 3.7% for female students) [2]. The high smoking prevalence among out-of-school adolescents was related to frequent interactions with peer groups who smoke away from school and home surveillance and to the relatively high accessibility of tobacco compared to that of students in schools [3,4].

Smoking started in adolescence is characterized by a continuous increase compared to smoking started in adulthood due to increased nicotine tolerance as the duration of smoking increases [5]. Smoking during adolescence increases the likelihood of smoking in adulthood, and the higher the amount of smoking during adolescence, the higher the risk of adult smoking [6]. The amount of smoking is a variable that is directly related to the risk of tobacco, and to be protected from the harmful effects of tobacco, it is necessary to control the amount of smoking from the time of adolescence. According to a study on the health condition of out-of-school youth and health promotion strategies, the smoking rate of heavy users who smoke more than 10 cigarettes a day was high: 57.9% for out-of-school adolescents and 19.6% for adolescents in school [7]. The amount of smoking in adolescence was associated with school dropout. The dropout rate of the group with the highest smoking rate was 44.4%, and the dropout rate of the group with the next-highest smoking rate was 35.6% [8]; thus, it can be inferred that out-of-school adolescents smoked more than those in school. Factors influencing adolescent smoking vary, including individual and socioenvironmental factors.

Nonetheless, it is crucial to grasp the socioecological dynamics of youth, such as the characteristics of individuals and their families, friends, and communities in which their social relationships develop, to understand the current risks and preventive factors [9]. Socioecological relationships define humans and their environment in a mutually comprehensive relationship and identify health behavior factors from a social perspective that can be used to develop multidimensional interventions and strategies [10]. Therefore, the aim of this study is to examine factors that affect tobacco use among adolescents from the perspective of a socioecological model of individual and interpersonal factors.

Previous studies have reported that individual cognitive factors, such as self-efficacy and self-control, influence whether adolescents continue or quit smoking. Adolescents with a high sense of self-efficacy showed a high rate of success in quitting [11], and the higher their self-control, the more effectively they resisted the temptation to smoke [12]. In particular, it was reported that out-of-school adolescents lack self-regulation ability and exhibit low self-control [13,14,15]. Several studies have confirmed that adolescents with low self-control have a higher smoking rate [16,17], and controversy has arisen with respect to the relationship between the amount of smoking and self-control in adolescents. In a study on adolescents attending schools, those with a high amount of smoking had low self-control [18], whereas in another study, the higher the self-control, the greater the amount of smoking [19].

A study by Madden and Johnson [20] revealed that smokers had lower self-control than non-smokers and that low self-control increased dependence on smoking, making smokers continue to smoke and making it difficult to quit [21,22]. On the other hand, it has been argued that if smokers think they can control their smoking behavior, they are more likely to smoke, and groups with high self-control have higher smoking rates, making it difficult to succeed in quitting [23].

In addition, it was found that the social network surrounding adolescents affects smoking [24,25]. A social network refers to “a network of social relationships surrounding an individual” [26]; for adolescents, social networks include friends and family, and adolescents start smoking by observing or taking these social contacts as models [27]. As adolescents age, they move away from the influence of their family and become more influenced by their friends [28]. Many previous studies have reported that the smoking of friends, rather than family members, has a greater effect on adolescent smoking [29,30]. The percentage of friends smoking and the social relationships formed among smoking friends are important factors affecting smoking [31]; therefore, it is necessary to understand the characteristics of the social networks of smoking among out-of-school adolescents and especially how friends’ smoking behavior influences the amount of smoking among adolescents.

To date, studies on out-of-school smoking youth have either dealt with smoking as a type of delinquency, along with various problematic behaviors [23,32], or with regard to the factors affecting smoking or re-smoking [33,34]. Only a few studies have observed specific smoking status and levels, such as the amount of smoking among out-of-school youth. The higher the amount of smoking in adolescence, the greater the amount of smoking in adulthood, which can have a detrimental effect on health, such as by causing lung cancer [35,36]. In particular, it was found that out-of-school adolescents smoke more than current students [31]; thus, it is necessary to elucidate relevant factors and interventions to reduce the amount of smoking among out-of-school youth.

From the ecological viewpoint, factors that affect the health behaviors of adolescents can be found in various environments, and they do not independently influence their behavior and development [26]. In short, the structure and environments that affect adolescents’ behavior and development are in continuous interaction [27]. Hence, interaction among major environmental factors of adolescents, such as family, school, and friends, as well as intrapersonal factors, affect individuals in various ways [27]. Based on an ecological model [37] of out-of-school adolescent smoking behavior, in this study, we aim to identify individual and interpersonal factors affecting tobacco use among out-of-school adolescents. We expect to provide basic data to help develop interventions that can reduce the amount of smoking among out-of-school adolescents and help them succeed in quitting.

The aim of this study is to identify factors that affect the amount of smoking among out-of-school adolescents. The specific objectives are as follows:Examine differences in the amount of smoking according to the general characteristics of out-of-school adolescent smokers;Examine the level of self-control, characteristics of social networks of friends, and amount of smoking among out-of-school adolescent smokers;Examine the correlation between self-control, characteristics of social networks of friends, and amount of smoking among out-of-school adolescent smokers; andExamine the effect of self-control and social networks of friends on the amount of smoking among out-of-school adolescent smokers.

## 2. Materials and Methods

### 2.1. Study Design

This study is a cross-sectional study with the aim of identifying the socioecological factors affecting tobacco use among out-of-school youth. Among these socioecological factors, self-control was selected as an individual factor, and social networks of friends was selected as an interpersonal factor to study the effects on tobacco use from a multidimensional perspective.

### 2.2. Participants

The subjects of this study were 187 out-of-school adolescent smokers. In this study, smokers were classified as such if they responded, “one or more cigarette a day” and “one or more day per month” to questions about the average amount of cigarettes smoked per day and the number of days smoking in the past 30 days. The study population comprised out-of-school youth from the J-province(Do) Youth Counseling Welfare Center as of 2020. The sample size was determined using G*Power3.1 software. We conducted a regression analysis with a medium effect size, a significance of 0.05, a power of 0.90, and a two-tailed test [34]. The minimum sample size was 158. A total of 190 surveys were collected considering the dropout rate, and only three surveys with insufficient responses were excluded; therefore, a total of 187 surveys were used for the final analysis. The target audience included those aged 13–18 who had not received regular middle and high school education in the past 6 months and had visited the J-do Youth Counseling Welfare Center at least once. This age group was selected as study subjects because the rate of school dropout and smoking among middle and high school students was relatively high. Following the guidelines of the institutional review board, in the case of minors, a parental consent form is essential; thus, adolescents who had submitted a parental consent form were included in the study. In cases in which youth wished to participate in the study but had not received consent from their guardian, the consent of a counselor at the Youth Counseling Welfare Center was obtained as a guardian’s representative (IRB approval number: WS-2022-06).

### 2.3. Measures

(1)General characteristics

The general characteristics of out-of-school youth included age, gender, time of school dropout, duration of school dropout, reason for school dropout, living with parents or not, average allowance, age at which they had started smoking, and duration of smoking. The time of school dropout was classified as ‘middle school’ or ’high school’; the duration of school dropout was classified as ‘less than 1 year’ or ‘1 or more years’; and the reasons for dropout were ’difficulty in studying’, ’lack of necessity of studying’, ‘violation of school rules’, ’family circumstances’, ’making friends outside of school’, and ‘conflict with parents’. The average allowance was ‘KRW 50,000 or less’ or ’more than KRW 50,000’. The age at which participants had started smoking was classified as ’age of 10–12′, ’age of 13–15′, or ’age of 16–18′; and the smoking period was classified as ’less than 1 year’, ’1 year to less than 2 years’, ‘2 years to less than 3 years’, or ’3 or more years’.

(2)Self-Control

To measure adolescents’ self-control, the self-control scale developed by Gottfredson and Hirschi [38] was used after being modified and supplemented by Nam and Ok [39]. This tool consists of 20 items, including 10 items about seeking long-term satisfaction and 10 items about seeking immediate satisfaction. Long-term satisfaction measures one’s ability to concentrate, think before acting, delay cravings, and solve problems effectively. Immediate satisfaction measures the tendency to be impulsive and self-centered and take action over words. The self-control scale is a 5-point Likert scale ranging from 1 (*not at all*) to 5 (*strongly agree*). The total score ranges between 20 and 100, with higher scores indicating higher self-control. The Cronbach’s α value of Nam and Ok’s [39] scale was 0.82. In this study, Cronbach’s α was 0.88.

(3)Social networks of friends

To measure social networks related to adolescent smoking, Longabaugh and Zywiak’s [40] Smoking Important People Instrument was used after translation and modification. This tool consists of 10 items; however, as one item relates to the effect of smoking cessation treatment did align with the purpose of this study, a total of nine items were used. Question 1 asked the respondents to list as many as 10 people who were important in their social network; question 2 asked participants to indicate the relationships among the listed important people; questions 3–9 asked participants to provide information related to the listed important people; questions 3, 6, and 7, which measured the contact frequency, smoking amount, and smoking frequency of social network members, were measured as continuous variables; questions 4, 5, 8, and 9 measured the ties of social network members, the frequency of social support, support for smoking behavior, and support for quitting behavior. Questions 4, 5, 8, and 9 were rated on a 5-point Likert scale ranging from 1 (*not at all*) to 5 (*always*). In this study, only data on friends were extracted and used among important people. This tool measured the percentage of friend smokers, friend smoking support, friend smoking cessation support, and penetration of smoking friends. The formula for calculating the penetration of smoking friends was as follows.

  (3.1)Percentage of smoking friends

The percentage of smoking friends was calculated as the percentage of friend smokers among members of the participant’s social network [40].

  (3.2)Support by friends for smoking and smoking cessation

Support for smoking and smoking cessation was calculated by adding up the degree of support for smoking and smoking cessation among friends in social networks (on a 5-point scale), and the total score ranged from 1 to 50. The degree of support for smoking and smoking cessation ranges from 1 (*not at all*) to 5 (*strongly agree*) [40]. The higher the total score of support for smoking behavior, the greater the support for smoking among members of social networks. The higher the total score for smoking cessation support, the greater the support for smoking cessation among social network members.

  (3.3)The involvement of friend smokers in social networks

The involvement of friend smokers in the social network refers to smokers in the social network, friends and family who smoke, and the degree of smoking infiltration by members in the social network. Smokers in the social network are the smokers and friends who smoke represented as a percentage of the total number of people in the social network. The degree of smoke involvement by members of the social network is the sum of the contact frequency (times/day) of each member of the social network multiplied by their smoking intensity (number of cigarettes per day) [40].

(4)Amount of smoking

“How many cigarettes do you smoke on average per day?”. The average number of cigarettes smoked per day was measured with open-ended responses for the previous 30 days [40].

### 2.4. Data Collection

The data used in this study were collected from 15 February to 31 March 2020 from youth who visited the J-Do Youth Counseling Welfare Center. A notice on the recruitment of research subjects was attached to the bulletin board of the Youth Counseling Welfare Center, and the teacher in charge distributed questionnaires, parental consent forms, and written consent forms to the youth who expressed their intention to participate in this study. In the case of out-of-school adolescents who consented to participate in the study but did not have a guardian, consent was obtained by proxy through the Youth Counseling and Welfare Center teacher. Once the surveys had been completed, they were sealed in a paper bag and placed in a collection box designated by the center. The researcher collected the survey questionnaires through the collection box.

### 2.5. Data Analysis

The collected data were analyzed using the SPSS 25.0 program. Statistical significance was set based on the significance level of 5% and analyzed. The general characteristics of out-of-school adolescent smokers were analyzed by frequency and percentage, and the difference in smoking amount according to general characteristics was analyzed by mean, standard deviation, *t*-test, and ANOVA (analysis of variance). The self-control, characteristics of social networks of friends, and smoking amount of out-of-school adolescent smokers were analyzed by mean, standard deviation, skewness, and kurtosis. Correlation analysis was conducted on the relationship between self-control, characteristics of friends in the social network, and the amount of smoking among out-of-school adolescent smokers. Hierarchical regression analysis was performed to understand the effects of self-control and social networks of friends on the amount of smoking among out-of-school adolescent smokers.

## 3. Results

### 3.1. Difference in the Amount of Smoking According to the General Characteristics of Out-of-School Adolescents

The difference in the amount of smoking among out-of-school adolescents according to the subjects’ general characteristics is reported in Table 1. The amount of smoking was significantly higher for those aged 16 and older than those aged 15 and younger (*p* < 0.001) and significantly higher for those who did not live with their parents than for those living with their parents (*p* = 0.047). In addition, the amount of smoking among out-of-school adolescents was significantly higher for those with an average allowance of KRW 50,000 or more than those with less an allowance of less than KRW 50,000 (*p* = 0.001).

### 3.2. Out-of-School Adolescents’ Self-Control, Social Networks of Friends, and Amount of Smoking

The self-control and social networks of study subjects, namely out-of-school adolescents, are reported in Table 2. Self-control scores were 2.61 ± 0.38 points, and the percentage of friends smoking in their social networks was 82.89%. Smoking support was 4.32 ± 3.27 points, support for smoking cessation was 4.32 ± 3.27 points, the degree of penetration of smoking friends was 318.65 ± 354.89, and the smoking amount was 7.51 ± 5.71 cigarettes. The skewness of self-control, traits, and actions of friends in social networks was less than ±2, and kurtosis was less than ±7, satisfying the standard of normality assumption [41].

### 3.3. Correlation between Self-Control, Social Network of Friends, and Amount of Smoking among Out-of-School Adolescents

Prior to regression analysis, Pearson’s correlation analysis was performed to confirm the one-to-one correlation between continuous variables. Table 3 shows the results of analyzing the correlation between the amount of smoking, self-control, and variables in the social network of friends among out-of-school adolescents. Smoking amount showed no correlation with self-control and variables in the social network of friends, but self-control had a negative correlation with smoking support by friends (r = −0.178, *p* = 0.018), a positive correlation with cessation support by friends (r = 0.234, *p =* 0.002), and a negative correlation with the smoking of friends (r = −0.196, *p* = 0.009). This shows that as self-control increases, friends’ support for smoking decreases, and their support for quitting smoking increases. The higher the smoking rate of friends, the lower the smoking involvement of respondents.

The smoking rate of friends showed a positive correlation with cessation support by friends (r = 0.471, *p* < 0.001), a negative correlation with smoking support by friends (r = −0.184, *p* = 0.014), and a positive correlation with smoking penetration of friends (r = 0.729, *p* < 0.001). A positive correlation was observed between smoking support by friends and smoking penetration of friends (r = 0.339, *p* < 0.001).

As friends quit smoking, their support for smoking increases. The higher the smoking rate of friends, the lower their support for smoking cessation. The higher the smoking rate of friends, the higher their smoking involvement. The higher the support for smoking from friends, the higher the degree of smoking involvement. 

### 3.4. Influence of Self-Control and Social Networks of Friends on the Amount of Smoking among Out-of-School Adolescents

The analysis results of factors affecting the amount of smoking among out-of-school adolescents are reported in Table 4. Prior to the analysis, general characteristics with significant differences in terms of smoking amount, such as age, living with parents or not, and average allowance, were treated as dummy variables in the first step, and self-control and characteristics of the social network were added step by step. As a result of checking the multicollinearity between independent variables in this study, the variance inflation factor (VIF) was greater than or equal to 1 and less than or equal to 10, indicating no problem.

Model 1, in which the abovementioned general characteristics were added as control variables, was significant (F = 16.36, *p* < 0.001). The significant variables were age (β = 0.257), living with parents (β = −0.267), and average allowance (β = 0.260). In other words, the higher the age, the lower the ratio of living with parents; and the higher the average allowance, the higher the smoking amount. The explanatory power of the variables for smoking amount was 20.8%.

Model 2, to which self-control was added, was significant (F = 12.20, *p* < 0.001), and the explanatory power of smoking amount was 20.4%. The significant variables in model 1 were age, living with parents, and average allowance, whereas self-control was not significant. Model 3, in which self-control and characteristics of the social network of friends were added, was significant (F = 8.77, *p* < 0.001). The significant variables in model 3 were age, living with parents, and average allowance, which were the same as in model 1 and model 2. The smoking rate of friends (β = 0.256) and the degree of penetration of friends smoking (β = 0.341) were significant variables. The higher the percentage of friends smoking and the higher the smoking penetration of the members of social networks, the higher the smoking amount. The explanatory power of the variables for the smoking amount was 26.2%.

## 4. Discussion

In this descriptive research study, we examined the socioecological factors affecting tobacco use among out-of-school adolescents. Self-control was considered as an individual factor, and social networks of friends was considered as an interpersonal factor to collect data for the development of a multidimensional intervention for out-of-school youth smoking. First, according to the general characteristics of out-of-school adolescents, the amount of smoking was higher among adolescents aged 16 and older than among adolescents aged 15 and younger, as well as among those who did not live with their parents compared to adolescents living with their parents. Adolescents aged 16 years and older had a longer smoking period, which may lead to an increase in nicotine dependence and amount of smoking [42]. These results are consistent with the results of studies reporting an increased risk of daily smoking among children not living with their parents [43].

The amount of smoking was significantly higher among adolescents with an average allowance of KRW 50,000 or more compared to those with an average allowance of less than KRW 50,000, which supports the results of a previous study showing that the greater the allowance, the higher the amount of smoking [14]. According to a study investigating the factors affecting the amount of smoking among adolescents attending school [44], high school students smoked more than middle school students, and adolescents with high household economic status smoked more than those with low household economic status. In addition, the risk of heavy smoking (i.e., smoking 20 or more cigarettes per day) increased by 3.78 times in adolescents who did not live with their families compared to adolescents living with their families. This indicates that the amount of smoking among adolescents attending school was also related to age, living with parents, allowance, and household economic status, as in out-of-school adolescents.

The self-control score of out-of-school adolescent smokers was 2.61—lower than the median level. The percentage of friends smoking was 82.89%, and the number of cigarettes smoked was 7.51. In a study on the amount of smoking using the health behavior survey data for adolescents in schools, the self-control score of the smokers was 66.73 points (3.34 points converted to a five-point scale), which was slightly higher than the median level; however, we could not directly compare the scores due to the use of different research tools [44]. In this study, the self-control score of out-of-school adolescents was relatively low, which was consistent with the results of a previous study [45] reporting that the higher the self-control of adolescents, the lower the dropout rate. In a study on the relationship between low self-control and smoking in adolescence, the problem of self-control was indicated as the cause of adolescent smoking [13,17]; therefore, it is necessary to continuously monitor the relationship between smoking and self-control among out-of-school adolescents.

In a study on middle school students reporting on smoking in school [19], the rate of friends smoking was 84.6%, which was similar to the results reported in this study, and the smoking amount of adolescents in schools was 11.83 cigarettes, which higher than that reported for adolescent smokers in this study. However, in a nationwide survey on the health of out-of-school youth, the proportion of out-of-school adolescents who smoked 10 or more cigarettes a day was 57.9%—higher than the 19.6% of students reported in schools [7].

In this study, the percentage of smoking friends and the smoking penetration of friends in the social network were identified as factors affecting the amount of smoking among out-of-school adolescents. Interpersonal factors, such as social network of friends, were confirmed to affect tobacco use among out-of-school youth, indicating that the higher the percentage of friends who smoked, the higher the amount of smoking. In a study of adolescents in school, the daily smoking rate was 17.82 times higher when a friend smoked, and the risk of smoking 20 cigarettes or more a day increased by 5.54 times [6,45,46]. In addition, in this study, the higher the smoking penetration of friends, the higher the smoking amount. Friends’ smoking penetration is a variable that reflects the frequency of contact with friends and the amount of smoking with friends. It can be interpreted that the more frequent one’s communication with friends and the more these friends smoke, the more one smokes. These results are similar to those reported in a study on adolescents in schools that found that friends’ smoking has a greater influence on adolescent smoking that parents’ smoking in adolescents’ social networks [47,48].

The results of this study reflect the characteristics of adolescent smoking behavior. Most adolescents start smoking under the influence of friends [29,33], and once they start smoking, they develop strong bonds [30]. In particular, it can be inferred that smoking adolescents have friends who smoke in their social networks and socialize and live with them. This leads to an increase in the frequency of smoking, and therefore, the amount of smoking increases. However, if a significant proportion of members of the social network of adolescents oppose smoking, the amount of smoking may decrease, and their smoking behavior may improve. This is supported by the finding that social support for quitting smoking positively impacts smoking cessation in adolescents [49].

For adolescents, smoking is an unhealthy behavior transmitted and persisting in social networks [45,46]. In terms of understanding adolescents’ smoking behavior, understanding the characteristics of social networks is just as important as understanding the psychological characteristics of individuals related to smoking [47,48]. Therefore, smoking cessation counseling should consider the individual characteristics of smoking adolescents, who should seek group interventions to reduce their amount of smoking and succeed in quitting smoking with friends by accessing the social networks of smoking adolescents. In addition, a policy intervention should increase the number of smoking cessation counselors, friends, and family who provide social support for smoking cessation and recommend quitting smoking in the social network of out-of-school adolescents.

In this study, we found that out-of-school adolescents’ self-control did not affect the amount of smoking. There are conflicting views on the relationship between smoking amount and self-control in adolescents. Low self-control increases dependence on smoking, which makes it difficult to quit smoking [21,22]. However, if smokers believe they can control their smoking behavior because they have high self-control, they are more likely to perceive smoking more positively, which leads to a high smoking rate [23]. Therefore, the relationship between self-control and the amount of smoking needs to be confirmed continuously through more follow-up studies.

Through this study, we investigated self-control and social networks of friends with respect to the amount of smoking among out-of-school adolescents. As the amount of smoking during adolescence increases, the amount of smoking in adulthood increases, leading to more fatal health risks, such as lung cancer, in adulthood [35]. The significant variables in model 3 were age, living with parents, and average allowance, which were the same as in models 1 2. The smoking rate of friends (β = 0.256) and the degree of penetration of friends smoking (β = 0.341) were significant variables. The higher the percentage of friends smoking and the higher the smoking penetration of the members of social networks, the higher the smoking amount. It is expected that intervention programs that consider socioecological factors influencing tobacco use will be developed based on this study for out-of-school youth to help them quit smoking. 

The limitations of this study are as follows. First, in this cross-sectional study, we analyzed the relationship of factors affecting the amount of smoking among out-of-school adolescents; however, the collected data cannot explain the causal relationship between variables. Second, in this study, we analyzed only the individual and interindividual factors of the ecological model and could not confirm the relevance of community factors and policy factors. Third, this study is subject to limitations with respect to generalization because we selected and investigated participants from a limited area.

## 5. Conclusions and Recommendations

This study confirmed that interpersonal factors, among socioecological factors, affect the tobacco use of out-of-school youth. Friends’ smoking rate and smoking penetration rate variables were shown to have positive effects: the higher the friends’ smoking rate and penetration rate, the higher the tobacco use in adolescents. Hence, this study proposes the following:(1)We found that the higher the percentage of friends smoking and the higher the smoking penetration of friends, the higher the amount of smoking. To reduce the amount of smoking among out-of-school adolescents and ultimately lead them to quit smoking, appropriate leisure activities should be provided to adolescents within the social network of friends who smoke so that smoking does not become a medium for maintaining social relationships. In addition, smoking cessation interventions should be provided individually and within the meaningful social network of friends surrounding the individual.(2)This study is subject to limitations with respect to generalizability of the study results when extracting subjects from a random sample. A survey with a larger, nationwide sample of out-of-school youth would increase the generalizability of the results of subsequent studies.(3)To develop successful preventive measures against tobacco use among youth, education and counseling programs based on socioecological characteristics should be considered, and strong tobacco sales regulations that limit adolescents’ access to tobacco should be implemented.

## Figures and Tables

**Table 1 healthcare-10-02138-t001:** Amount of smoking according to the general characteristics of out-of-school adolescents (*N* = 187).

Variable	Category	*n* (%)	Mean ± SD	*p*
Age	≤15	77 (41.2)	5.49 ± 2.48	<0.001
	≥16	110 (58.8)	8.93 ± 6.81	
Gender	Men	134 (71.7)	7.69 ± 5.89	0.321
	Women	53 (28.3)	7.08 ± 5.24	
Discontinuation of school (time)	Middle school	113 (60.4)	7.00 ± 4.62	0.415
	High school	74 (39.6)	8.30 ± 7.01	
Discontinuation of school (period)	<1 year	54 (28.9)	7.87 ± 7.13	0.491
	≥1 years	133 (71.1)	7.37 ± 5.04	
Reasons for discontinuation of school (multiple responses)	Difficulty studying			
	No	100 (53.5)	8.24 ± 6.81	0.158
	Yes	87 (46.5)	6.68 ± 3.96	
	Lack of study needs			
	No	98 (52.4)	7.06 ± 4.84	0.379
	Yes	89 (47.6)	8.01 ± 6.52	
	Violation of school regulations			
	No	142 (75.9)	7.61 ± 6.12	0.577
	Yes	45 (24.1)	7.22 ± 4.16	
	Economic status of family			
	No	154 (82.4)	7.86 ± 6.13	0.085
	Yes	33 (17.6)	5.91 ± 2.47	
	Out-of-school friends interaction			
	No	128 (68.4)	7.51 ± 5.45	0.916
	Yes	59 (31.6)	7.53 ± 6.27	
	Conflict with parents			
	No	137 (73.3)	7.43 ± 4.94	0.610
	Yes	50 (26.7)	7.74 ± 7.46	
	Etc.			
	No	186 (99.5)	7.53 ± 5.72	0.727
	Yes	1 (0.5)	5.00 ± 0.00	
Live with parents	No	6 (3.2)	20.67 ± 14.51	0.047
	Yes	181 (96.8)	7.08 ± 4.67	
School achievement	High, medium	60 (32.1)	7.48 ± 6.46	0.228
	Low	127 (67.9)	7.53 ± 5.34	
Allowance (KRW ten thousand)	<5	117 (62.6)	6.67 ± 5.34	0.001
	≥5	70 (37.4)	8.93 ± 6.05	
Start smoking (age)	10–12	29 (15.5)	5.38 ± 2.16	0.103
	13–15	112 (59.9)	7.80 ± 6.41	
	16–18	46 (24.6)	8.15 ± 5.17	
Smoking period (years)	<1	53 (28.3)	6.15 ± 4.60	0.089
	1–2	24 (22.5)	7.67 ± 4.64	
	2–3	54 (28.9)	7.96 ± 5.57	
	≥3	38 (20.3)	8.61 ± 7.83	

**Table 2 healthcare-10-02138-t002:** Self-control, traits, and actions of friends in social networks, as well as amount of smoking among out-of-school adolescents (*N* = 187).

Characteristics	Range	Mean ± SD	Skewness	Kurtosis
Self-control	1–5	2.61 ± 0.38	−0.24	0.97
Social networks traits				
Friends smoking	0–100	82.89 ± 32.03	−1.67	1.45
Friends smoking support	1–40	6.51 ± 3.62	4.45	40.43
In) Friends Smoking support	0–3.71	1.93 ± 0.42	−0.28 ^(1)^	1.52 ^(1)^
Friends No-smoking support	1–30	4.32 ± 3.27	4.39	29.38
In) Friends No-smoking support	0–3.43	1.56 ± 0.45	0.72 ^(1)^	1.42 ^(1)^
Social network friend smokers’ involvement	0–2464	318.65 ± 354.89	2.95	11.99
ln) Social network friend smokers’ involvement	0–7.81	5.16 ± 1.41	−1.83 ^(1)^	5.02 ^(1)^
Daily smoking amount	1–40	7.51 ± 5.71	2.64	9.29
ln (Daily smoking amount)	0.69–3.71	1.98 ± 0.55	0.34 ^(1)^	0.70 ^(1)^

^(1)^ Values are represented as log transmission variables due to high skewness and kurtosis.

**Table 3 healthcare-10-02138-t003:** The correlation between self-control, traits of friends in social networks, and smoking amount among out-of-school adolescents (*N* = 187).

	Y	X1	X2	X3	X4	X5
	r (*p*)					
Y: Daily Smoking Amount	1					
X1: Self-control	0.136 (0.064)	1				
X2: Friends smoking (%)	−0.059 (0.419)	−0.104 (0.155)	1			
X3: Friends’ smoking support	0.000 (0.995)	−0.178 (0.018)	0.471 (<0.001)	1		
X4: Friends’ smoking cessation support	0.013 (0.862)	0.234 (0.002)	−0.184 (0.014)	0.072 (0.341)	1	
X5: Smoking involvement among social network friends	0.114 (0.129)	−0.196 (0.009)	0.729 (<0.001)	0.339 (<0.001)	−0.97 (0.200)	1

**Table 4 healthcare-10-02138-t004:** Factors influencing the smoking amount of out-of-school adolescents (*N* = 187).

Characteristic	Model Ⅰ			Model Ⅱ			Model Ⅲ		
	β	t	*p*	β	t	*p*	β	t	*p*
Age	0.257	3.801	0.000	0.258	3.645	0.000	0.294	3.998	0.000
Live with parents	−0.267	−3.953	0.000	−0.267	−3.939	0.000	−0.275	−4.178	0.000
Allowance	0.260	3.867	0.000	0.261	3.796	0.000	0.225	3.274	0.001
Self-control				−0.002	−0.028	0.978	0.060	0.838	0.403
Traits of social networks of friends’									
Friends smoking (%)							0.256	2.518	0.013
Friends’ smoking support							0.133	1.668	0.097
Friends’ smoking cessation support							−0.110	−1.468	0.144
Smoking involvement among social network friends							0.341	3.670	0.000
F	16.36(<0.001)			12.20(<0.001)			8.77(<0.001)		
R^2^	0.222			0.222			0.296		
Adjusted R^2^	0.208			0.204			0.262		

## Data Availability

Not applicable.

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
