# Peer review of "Impact of Self-Control and Social Network of Friends on the Amount of Smoking among Out-of-School Youth"

_healthcare, 2022, doi:10.3390/healthcare10112138_

Round 1
Reviewer 1 Report (Previous Reviewer 2)
The authors have revised the manuscript for the first review comments. However, there are still some questions that the authors need to answer. I again raise review comments on this manuscript.
1. In line 28, I still suggest that the authors add the name of the country that enacted this law, because different countries have different laws. Even if it is a United Nations law, it must be stated that it is formulated by the United Nations.
2. In line 203-214, the "Purpose" section should be merged into one paragraph of "Introduction", because the purpose of the study is what the "Introduction" section should state. In addition, stating the purpose of the study in dots is not a good writing strategy. Authors should describe the purpose of the study in one paragraph.
3. In line 221-223, this paragraph is not a statement of research hypothesis. Especially lines 222-223, this is not how the research hypothesis is written, but refers to the research question. Research hypotheses need to be based on past research and theoretical inferences.
4. In line 234, the insufficient sample size of this manuscript remains a significant flaw. Although the authors present evidence to demonstrate the calculation of the minimum sample size, the authors ignore the limitations of analytical tools. The authors use SPSS as the analysis tool, which should meet the sample requirements of SPSS in regression analysis. When the sample size is insufficient, research results can be unstable. This would reduce the credibility of the findings.
5. In line 303-314, this paragraph is redundant. This adds a lot of noise to the manuscript. For readers, these descriptions are meaningless.
6. In line 317-482, the authors just removed the subtitle number without much correction. I suggest that the author reformulate the expression of this paragraph.
7. In the previous review comments, I raised the question of adopting a score-summation analysis strategy. While the authors responded, one of the study purposes was comparison with other studies. If the calculation methods are different, the comparative analysis will be difficult. This would reduce the contribution of this manuscript. Therefore, adopting the average strategy may be a more appropriate analysis method.
8. Has the formula for In line 472-478, penetration of smoking friends been used in past research? Or is this an innovation by the authors? If past research has been used, please include citations.
9. In line 524-535, the description of this statistical analysis needs to be restructured. Many basic statistical principles and methods do not need to be described in lengthy words. But the authors lack evidence to justify the use of hierarchical regression. In other words, please present literature evidence that hierarchical regression is appropriate in this study.
10. In the previous review comments, I put forward: "In Table 3, in both model 2 and model 3, self-control is not significant. This is a different result from past research. Unfortunately, the author did not discuss this Reason. Authors please add." The author did not respond to this review comment. Please add on.
11. In the “Discussion” section, the authors seem to have little specific research inspiration beyond validating past research. I suggest authors think carefully about the value of this manuscript and describe it in the "Discussion" section.
12. In line 942-955, authors are asked to abandon the column-point writing strategy and use the paragraph-based writing strategy instead.

Author Response
|
In line 28, I still suggest that the authors add the name of the country that enacted this law, because different countries have different laws. Even if it is a United Nations law, it must be stated that it is formulated by the United Nations. |
According to your comment, we added the name of the country that made the law. <p.1 line 28, 30> |
|
In line 203-214, the "Purpose" section should be merged into one paragraph of "Introduction", because the purpose of the study is what the "Introduction" section should state. In addition, stating the purpose of the study in dots is not a good writing strategy. Authors should describe the purpose of the study in one paragraph. |
Thank you for your kind comments. According to your comment, we have describe the purpose of the study in one paragraph. <p.2 line 107-115> |
|
In line 221-223, this paragraph is not a statement of research hypothesis. Especially lines 222-223, this is not how the research hypothesis is written, but refers to the research question. Research hypotheses need to be based on past research and theoretical inferences. |
Based on the reviewer's opinion, we acknowledge that the research hypothesis description is incorrect. Therefore, we will delete the research hypothesis.
|
|
In line 234, the insufficient sample size of this manuscript remains a significant flaw. Although the authors present evidence to demonstrate the calculation of the minimum sample size, the authors ignore the limitations of analytical tools. The authors use SPSS as the analysis tool, which should meet the sample requirements of SPSS in regression analysis. When the sample size is insufficient, research results can be unstable. This would reduce the credibility of the findings. |
Considering the statistical significance through regression analysis and the effect size of the practical significance criterion for a sample of 187 people, I think that there is no difficulty in conducting the study with the size calculated through the G*Power 3.1.9.7 program. |
|
In line 303-314, this paragraph is redundant. This adds a lot of noise to the manuscript. For readers, these descriptions are meaningless. |
Thank you for your kind comments. According to your comment, we have excise this paragraph |
|
In line 317-482, the authors just removed the subtitle number without much correction. I suggest that the author reformulate the expression of this paragraph. |
Thank you for your kind comments. According to your comment, we have reformulate the expression. |
|
In the previous review comments, I raised the question of adopting a score-summation analysis strategy. While the authors responded, one of the study purposes was comparison with other studies. If the calculation methods are different, the comparative analysis will be difficult. This would reduce the contribution of this manuscript. Therefore, adopting the average strategy may be a more appropriate analysis method. |
The calculation method of the tool was implemented according to the original tool of the developer.
Reference) Longabaugh, R.; Zywiak, W. Important People Instrument; Center for Alcohol & Addiction Studies, Brown University: Providence,RI, USA, 1998
|
|
Has the formula for In line 472-478, penetration of smoking friends been used in past research? Or is this an innovation by the authors? If past research has been used, please include citations |
Penetration of smoking friends been used in past research.
We had include citations[40]. Reference) Longabaugh, R.; Zywiak, W. Important People Instrument; Center for Alcohol & Addiction Studies, Brown University: Providence,RI, USA, 1998 |
|
In line 524-535, the description of this statistical analysis needs to be restructured. Many basic statistical principles and methods do not need to be described in lengthy words. But the authors lack evidence to justify the use of hierarchical regression. In other words, please present literature evidence that hierarchical regression is appropriate in this study |
In hierarchical regression, independent variables are sequentially analyzed according to the needs to be reviewed first in the study. Psychological, social, and environmental factors affect adolescent smoking behavior through a complex interaction. Based on an ecological model (McLeroy et al., 1988) of out-of-school adolescent smoking behavior, this study aims to identify factors related to smoking behavior between individuals as well as individual perspectives. Reference) McLeroy, K. R., Bibeau, D., Steckler, A., & Glanz, K. (1988). An ecological perspective on health promotion programs. Health Education & Behavior, 15(4), 351-377.
|
|
In the previous review comments, I put forward: "In Table 3, in both model 2 and model 3, self-control is not significant. This is a different result from past research. Unfortunately, the author did not discuss this Reason. Authors please add." The author did not respond to this review comment. Please add on. |
Thank you for your kind comments. According to your comment, we added discussion of self-control. <p.13, line 449-456> |
|
In the “Discussion” section, the authors seem to have little specific research inspiration beyond validating past research. I suggest authors think carefully about the value of this manuscript and describe it in the "Discussion" section. |
Thank you for your kind comments. According to your comment, We tried to describe the implications of the findings. <p.13, line 428-435, 445-448> |
|
In line 942-955, authors are asked to abandon the column-point writing strategy and use the paragraph-based writing strategy instead. |
Thank you for your kind comments. According to your comment, we have revised. |

Reviewer 2 Report (New Reviewer)
INTRODUCTION
Line 30: Region/country should be referenced here, as policies vary.
Lines 33-40: What population does this survey study reference? Again, prevalence varies by region. Also, technically these percentages are prevalence estimates, not rates -- prevalence is not a rate.
METHODS
Line 123: This is a cross-sectional study and that should be stated explicitly. If this were purely descriptive, there would be no hypotheses.
Lines 168-195: It would be helpful to present a table or figure of the questions.
Line 220: What is the question? Does it ask cigarettes per day on average? On smoking days? Unclear how this data is then used for analysis.
Lines 244-248: Not clear why hierarchical regression was required, and if it is, for example, due to clustering, then standard correlation analysis might not be appropriate.
RESULTS
Please avoid reporting p-values in text without the means or percentages for comparison.
Table 1 is very difficult to read due to spacing problems; t/F statistics are not necessary; would help to be explicit about what the outcome is (i.e., “smoking amount” is not specific –“ average number of cigarettes per day would be better”)
Unclear why skewness and kurtosis are being reported in Table 2 when they are not mentioned in the text interpretation. What do these values mean for the reader? Interpretation should be provided in the text.
Lines 299-302: To what results are these referring to? This seems like a repeating of the prior paragraphs?
Table 3: Did the authors assess whether Pearson’s r was appropriate for these data? Unclear whether these are linear relationships, for example.
Table 4: What is the rationale for including both ‘friends smoking support’ and ‘friends no smoking support’ in the same model?
DISCUSSION
This section is generally difficult to read due to poor organization, and it offers little interpretation of this study’s results, other than repeating what was already in the results section. There is also no discussion of limitations.
Line 436: This language is too strong – a cross-sectional study cannot “confirm” an “affect”
Not clear how these specific recommendations are supported by the study’s results. For example, this study did not examine leisure activities, so how can one conclude that these would be a benefit to out-of-school youth?
Author Response
|
Line 30: Region/country should be referenced here, as policies vary. |
According to your comment, we added the name of the country that made the law.
<p.1 line 28, 30> |
|
Lines 33-40: What population does this survey study reference? Again, prevalence varies by region. Also, technically these percentages are prevalence estimates, not rates -- prevalence is not a rate. |
According to your comment, we have correct it. <p.1 line 33-40> |
|
Line 123: This is a cross-sectional study and that should be stated explicitly. If this were purely descriptive, there would be no hypotheses. |
Thank you for your kind comments. According to your comment, we have correct it.
This study is a cross-sectional study to identify the socioecological factors affecting tobacco use among out-of-school youth. |
|
Lines 168-195: It would be helpful to present a table or figure of the questions. |
Thank you for your kind comments. According to your comment, we add it as a appendix. |
|
Line 220: What is the question? Does it ask cigarettes per day on average? On smoking days? Unclear how this data is then used for analysis. |
Thank you for your kind comments. According to your comment, we have correct it
“How many cigarettes do you smoke on average per day? |
|
Lines 244-248: Not clear why hierarchical regression was required, and if it is, for example, due to clustering, then standard correlation analysis might not be appropriate. |
In hierarchical regression, independent variables are sequentially analyzed according to the needs to be reviewed first in the study. Psychological, social, and environmental factors affect adolescent smoking behavior through a complex interaction. Based on an ecological model (McLeroy et al., 1988) of out-of-school adolescent smoking behavior, this study aims to identify factors related to smoking behavior between individuals as well as individual perspectives. Reference) McLeroy, K. R., Bibeau, D., Steckler, A., & Glanz, K. (1988). An ecological perspective on health promotion programs. Health Education & Behavior, 15(4), 351-377.
|
|
Table 1 is very difficult to read due to spacing problems; t/F statistics are not necessary; would help to be explicit about what the outcome is (i.e., “smoking amount” is not specific –“ average number of cigarettes per day would be better”) |
Thank you for your kind comments. According to your comment, we have correct it. We have give detail explanation of the term.
Amount of smoking “How many cigarettes do you smoke on average per day? “ The average number of cigarettes smoked per day was measured with open-ended responses for the previous 30 days[40].
|
|
Unclear why skewness and kurtosis are being reported in Table 2 when they are not mentioned in the text interpretation. What do these values mean for the reader? Interpretation should be provided in the text. |
Thank you for your kind comments. According to your comment, we have correct it Self-control, friends social networks traits actions Skewness was less than ±2 and Kurtosis was less than ±7, which satisfied the standard of normality assumption[41].
|
|
Lines 299-302: To what results are these referring to? This seems like a repeating of the prior paragraphs? |
Thank you for your kind comments. According to your comment, we have excise it
|
|
Table 3: Did the authors assess whether Pearson’s r was appropriate for these data? Unclear whether these are linear relationships, for example. |
Prior to regression analysis, Pearson’s correlation analysis was performed to confirm the one-to-one correlation between continuous variables |
|
Table 4: What is the rationale for including both ‘friends smoking support’ and ‘friends no smoking support’ in the same model? |
‘Friends smoking support’ and ‘friends no smoking support’ are a contradictory concept. Because the influence of friends who induce smoking and friends who recommend smoking are different, we put them both.
|
|
This section is generally difficult to read due to poor organization, and it offers little interpretation of this study’s results, other than repeating what was already in the results section. There is also no discussion of limitations. |
Thank you for your kind comments. According to your comment, we have correct it
We tried to describe the implications of the findings. < line 428-435, 443-448, 449-456>
We added the discussion of limitations. <Line 468-474> |
|
Line 436: This language is too strong – a cross-sectional study cannot “confirm” an “affect” |
Thank you for your kind comments. According to your comment, we have correct it
Through this study, we investigated self-control and the social network of friends as related the amount of smoking among out-of-school adolescents |
|
Not clear how these specific recommendations are supported by the study’s results. For example, this study did not examine leisure activities, so how can one conclude that these would be a benefit to out-of-school youth? |
Thank you for your kind comments. According to your comment, we have correct it <Line 443-448> |

Round 2
Reviewer 1 Report (Previous Reviewer 2)
This manuscript has been significantly improved in readability and academic value after two revisions. Although the small sample is still an issue in this manuscript, it is complemented by other statistics and discussions. I therefore suggest that this article be accepted in its current form.
This manuscript is a resubmission of an earlier submission. The following is a list of the peer review reports and author responses from that submission.
Round 1
Reviewer 1 Report
Thank you for the opportunity to review this manuscript. Overall, the manuscript addresses interesting questions. The manuscript focused on the influence of self-control and social network of friends on amount of smoking among out-of-school youth. A total of 187 adolescent smokers participated in this study, and surveys were used. However, there are several major problems with the study such as lack of a reasonable theoretical framework, and poor writing according to the standard.
First, this research is only a descriptive study, which lacks necessary theoretical significance. And as a descriptive study, the number of subject samples is relatively small, and the regional distribution of subject samples is single, so the generalization of conclusions is limited.
Second, the influence of self-control and social networking on youth smoking has been demonstrated in previous research, and the current study does not seem to propose new ideas and theoretical models.
Third, there are many non-standard formats and errors in the manuscript.
Reviewer 2 Report
Smoking in out-of-school adolescents is discussed in this manuscript. This is a health issue that needs attention. However, I still make the following review comments based on academic requirements.
1. Overall, smoking factors in adolescents have been frequently discussed in past studies. I cannot understand the research breakthrough and inspiration of the authors' manuscript. The authors should clearly state the value of this manuscript. Furthermore, the authors should discuss each phenomenon in depth. Finally, the statistical analysis strategy should also be revised. It is necessary for the authors to make substantial manuscript revisions or to rewrite the manuscript. Next, I offer the following comments on the details of the manuscript.
2. In “Abstract”, I make the following suggestions.
(1) In line 8, the authors indicate that this manuscript is "a descriptive study". This term needs to be revised. The authors used three methods to analyze the data, including t, ANOVA, and regression. All three methods should be classified in the category of inferential statistics. In general, descriptive research is only a primary data analysis. It is difficult for authors to make in-depth academic research based solely on descriptive statistics, unless qualitative research methods are employed. In other words, this sentence is not only wrong but also redundant. The same error also appears at line 117. I suggest that the authors correct them.
(2) In general, it is necessary to state the research method in the "Abstract". But because the authors' statistical methods for analyzing the data fall within the scope of basic statistics, the presentation in "Abstract" instead highlights the weakness of the manuscript. I suggest that authors may consider removing the description of statistical methods.
(3) In line 14-18, this paragraph is to illustrate the findings of this manuscript. However, this way of writing is difficult for readers to understand. In fact, Models 1-3 refer to the same research proposition. Instead, the authors' descriptions have the effect of being confounding. This practice is not encouraged. I suggest that authors rethink the way this paragraph is presented.
(4) In line 18-23, the authors state the study suggests in this paragraph. This suggestion is not based on data from the study results, but rather the authors' imagination. Therefore, it is inappropriate to write such a description in "Abstract". Please correct it.
In general, I suggest that authors rewrite the "Abstract" content.
3. In line 31, the authors refer to "Act on the support for out-of-school youth". I suggest that the authors add the name of the country that enacted the law (I guess South Korea) as each country has a different legal opinion on this. Also, I would like to know if there is an age limit for adolescents to buy cigarettes in South Korea? If such a law exists, authors may discuss the implications and limitations of this law in the appropriate section of the manuscript.
4. In line 104-114, it is not a good strategy for authors to take a column-point statement of research purpose. I suggest that authors briefly organize the purpose of this manuscript and state it in paragraphs. If the authors accept my suggestion, the "1.1" and "1.2" subtitles can be removed.
5. The authors analyzed the data with different statistical methods, including t, ANOVA, and regression. Statistical methods are tools used to solve problems, but only with clear hypotheses. It is a pity that the authors did not present the research hypotheses and the research inferences required for the hypotheses in the manuscript. I suggest that authors can supplement the manuscript with explanations and inferences of the hypothesis.
6. In the “Methods” section, I make the following suggestions.
(1) Authors have too many subsections in this section. I suggest that authors can consolidate subsections with less content.
(2) In line 117-118, this text states that it is not necessary to be a subsection by itself. This paragraph can be incorporated into other subsections.
(3) In line 119, I suggest changing the subtitle to "Participants".
(4) In line 127-128, the authors describe as follows:
“As a result, 158 subjects were calculated (Faul, Erdfelder, Buchner & Lang, 2009).”
I can't understand what the authors meant by writing this sentence. I would like to know if the sample size for this manuscript is 187 or 158.
(5) Regardless of whether the number of samples collected by the authors is 187 or 158, the number of samples is insufficient for the requirements of regression analysis (using SPSS as the analysis tool). Insufficient samples can lead to unstable analysis results. In other words, the analysis results are questioned. I suggest that authors should consider other statistical tools, such as VB-SEM (PLS-SEM), if they still want to take a statistical approach to regression analysis.
(6) In line 141-152, this text is redundant, because the relevant information can be found in Table 1.
(7) The authors have adopted a score-summation strategy in the analysis of many closed-ended questionnaires (e.g., self-control, line 160-161). Although the statistical calculation by summing the scores of all the questions is the same as the result of the statistical analysis after averaging the scores of all the questions, the authors are still advised to adopt the latter analysis method in academic practice (average strategy). The advantage of this is that the results of this manuscript are easily comparable to other studies. Therefore, I suggest that authors can change the way the data is handled.
(8) In line 141-197, I suggest that authors do a consolidation of the content, because there are too many subtitles in this paragraph.
(9) In line 193-195, I cannot understand the calculation formula of Penetration of smoking friends. Has any past research ever used it? If not, please explain the basis of the formula design more clearly.
(10) In line 197, is the measurement of the amount of smoking closed-ended questionnaire or open-ended respondents?
(11) In subsect 2.5, I suggest that the author make a simplified description and integrate it into other subsections, because these statistical methods are common and basic common sense.
(12) In line 221-223, the authors' approach to hierarchical regression is inappropriate. First, it is wrong to blindly use demographic variables as control variables. This concept has been discussed in many scholarly articles of statistics over the past 10 years. If the authors insist on using the demographic variable as a control variable, the authors are asked to provide supporting references and describe the procedure for the analysis. Secondly, please explain the reasons why model 2 and model 3 deal with independent variables separately. Finally, the authors should also justify the reasoning that hierarchical regression is an appropriate method of analysis in small sample data. In line 277-281 is also a misuse of control variables, please correct them together.
7. In the “Results” section, I make the following suggestions.
(1) In line 231, the p value should be .047 instead of <.001 (live parents; Table 1).
(2) In line 233, the p value should be .001 instead of <.001 (allowance; Table 1).
(3) In Table 2, there are some variables whose skewness and kurtosis values exceed acceptable thresholds. Please explain the reasons for this phenomenon. According to the data results, I guess the author may not delete the outlier value. Authors should clean up the outlier values before performing statistical analysis.
(4) In Table 2, the correlation coefficient for X2 and Y is -.059. Please explain the reasons and significance of the negative relationship between these two variables. In addition, authors should note the meaning of the numbers (with and without parentheses) in each column of Table 2. Finally, please explain the purpose of analyzing the correlation coefficient of each variable.
(5) In Table 3, self-control is insignificant in both model 2 and model 3. This is a different result from past studies. It is a pity that the author did not discuss this reason. Authors please add.
(6) In line 276, the description of “ref) Live with parents” is redundant. Please delete it.
8. In the “Discussion” section, the authors’ statements are weak because it is merely a repeated statement of the phenomenon and an explanation of whether it is the same as past research. What this manuscript lacks is an in-depth discussion of the phenomenon, including theoretical breakthroughs and practical inspiration. I suggest that the authors discuss the findings and contributions of this study in more depth.
9. In the “Conclusion and Implications” section, authors should add a description of “Research limitations” and “Future research”.
10. Based on the rigor and consistency of academic articles, I have the following suggestions on the format.
(1) Please provide the English name of the first author.
(2) In line 31, the author is requested to revise the quotation marks (「」) into English usage (“”).
(3) In accordance with the format requirements of this journal, authors are requested to indicate all citations with numerical codes.
(4) In line 110, there is one less period at the end of the sentence.
(5) In line 122, "moer" should be corrected to "more".
(6) In line 195, one more space before “cigarette”. Please delete it.
(7) In line 231, ")." is redundant. Please delete it.
(8) The "p" symbol for all p values is lowercase and italicized. Please correct them (line 229, 231, 233, 255-261 and 282).
(9) All "n" symbols for sample size are lowercase and not italicized. Please correct them.
(10) Only the first letter of the Table name is uppercase, and the rest are lowercase (including the text in the table). Please correct them.
(11) According to the principle of unity, please change "self control" to "self-control".
(12) In Table 3, "Freinds" should be corrected to "Friends" (X2).
(13) In line 341, there is a 4-letter space in front of the sentence. Please delete it.
(14) Please make appropriate typesetting for the blank space in P. 5 and P. 9.
(15) In the “References” section, please make corrections according to the journal format requirements.
